# Fabrication of Porous Hydrophilic CN/PANI Heterojunction Film for High-Efficiency Photocatalytic H$_2$ Evolution

**Xiaohang Yang [1,\*], Yulin Zhang [2], Jiayuan Deng [2], Xuyang Huo [1], Yanling Wang [1] and Ruokun Jia [2,\*]**

[1]   Department of Biomedical Engineering, Jilin Medical University, Jilin 132013, China
[2]   College of Chemical Engineering, Northeast Electric Power University, Jilin 132012, China
\*   Correspondence: xhyang16@jlu.edu.com (X.Y.); jiaruokun@163.com (R.J.)

**Abstract:** The modulation of surface wettability and morphology are essential to optimize the photocatalytic H$_2$ evolution activity of graphitic carbon nitride (CN)-based photocatalysts. In this work, the porous hydrophilic CN/PANI heterojunction film was prepared via interfacial polymerization and loaded on a porous PCL substrate. The construction of the type-II CN/PANI heterojunction enabled an overall spectrum response and the efficient separation and transportation of photoexcited charge carriers. The fabricated CN/PANI solid-state film in comparison with its powder counterpart elevated the utilization efficiency and maintained the long-term stability of photocatalyst. The porous morphology and hydrophilic surface increased the surface area and enhanced the surface wettability, favoring water-molecule adsorption and activation. The as-prepared CN/PANI heterojunction film exhibited photocatalytic H$_2$ production activity up to 3164.3 μmol·h$^{-1}$·g$^{-1}$, which was nearly 16-fold higher than that of pristine CN (569.1 μmol·h$^{-1}$·g$^{-1}$).

**Keywords:** CN/PANI heterojunction; porous solid-state film; hydrophilic surface; photocatalytic H$_2$ evolution





## 1. Introduction

Photocatalytic H$_2$ evolution via water splitting—that is, the application of solar energy to drive water reduction—has developed over the past decades years into an appealing route for the green production of hydrogen energy. This technique, through which incident photons are adsorbed by semiconductor photocatalysts to generate charge carriers participating in redox reaction, conforms to the concept of green chemistry and could help achieve the sustainable development goals [1–3]. Its productivity is mainly determined by the utilization range of the visible light spectrum and the photoelectric conversion efficiency of the photocatalysts [4]. A wide range of semiconductor materials are applied for photocatalytic H$_2$ generation, primarily including metal-based oxides, sulfides, and nitrides [5,6]. Non-metallic based polymers are introduced for photocatalysis owing to their easy modification, tunable band structures, and satisfactory visible light response [7]. Among them, graphitic carbon nitride (CN) has rapidly developed into one of the most promising photocatalysts for H$_2$ evolution from water due to its visible light-harvesting capacity (λ < 460 nm), strong reduction ability and environmentally friendly characteristics [8]. However, there are drawbacks related to pristine CN, including an insufficient adsorption range of the visible light spectrum and the poor separation and migration of the photoexcited charge carriers [9]. Heterojunction fabrication offers a feasible strategy to overcome these problems by integrating another semiconductor photocatalyst with a suitable band structure and superior electronic properties [10,11].

Polyaniline (PANI), a conductive polymer with an extended π-conjugated system, possesses a high absorption coefficient in the visible light range and high mobility of charge carriers [12]. It is selected as a preferred candidate to fabricate CN-based heterojunctions attributed to its wide spectral response, appropriate energy band edge, and high conductivity [13]. Ge et al. synthesized the CN/PANI heterojunction via in situ deposition oxidative

polymerization. The obtained CN/PANI heterojunction showed enhanced photocatalytic $H_2$ evolution activity due to the effectively improved carrier separation [14]. Zhang et al. presented the PANI nanorod array grown on the CN nanosheet by cryogenic dilution polymerization. The as-prepared CN/PANI heterojunction boosted photocatalytic $H_2$ production attributed to enhanced charge carrier transportation [15]. However, these reported CN/PANI heterojunctions were generally developed as a suspended powder in the photocatalytic system. There are some shortcomings in the use of the powder heterojunctions, such as difficult separation and segregation after the reaction [16]. The large losses and low utilization efficiency induced by the particulate properties of the heterojunctions could significantly constrain the photocatalytic $H_2$ generation performance. Meanwhile, the aggregation of powder heterojunctions would reduce the surface active sites, leading to decreased photocatalytic activity.

The membrane immobilization of powder photocatalyst is beneficial for long-term maintenance for photocatalytic activity and stability [17]. Generally, photocatalyst precursors that are dispersed into a solvent could be cast into solid state films through various methods, including spin coating, inkjet printing, and spray coating [18]. The porous membranes have been recognized as a suitable substrate for powder photocatalyst immobilization [19]. Herein, porous PCL film was selected as the substrate to obtain porous heterojunction film. In the presence of aniline monomer and pristine CN, the CN/PANI heterojunction was synthesized by interfacial polymerization and loaded on the porous PCL substrate. The obtained porous CN/PANI film exhibited a high surface area and hydrophilicity, thereby exposing abundant reactive sites and enhancing the water molecule adsorption. As a result, the as-prepared porous hydrophilic CN/PANI heterojunction membrane showed a nearly 16-fold increase in photocatalytic $H_2$ generation rate compared to that of the pristine CN.

## 2. Results and Discussion

### 2.1. Morphology and Structure Characterization

The morphology of pristine CN, PANI, and the CN/PANI heterojunction film were observed by scanning electron microscopy (SEM). Pristine CN showed a two-dimensional layer-like architecture with a size range of several micrometers (Figure 1a). Bare PANI exhibited agglomerated particles with an irregular shape (Figure 1b). The CN/PANI heterojunction film deposited on the PCL substrate showed large numbers of macropores and a rough surface (Figure 1c,d). As a comparison, the SEM image of the film made of CN/PANI heterojunction powder was shown in Figure S1. No regular porous structures were observed on its surface.

The FTIR spectra of the pristine CN, PANI, and the CN/PANI film was shown in Figure 2a. For pristine CN, the peak located at 813 $cm^{-1}$ was assigned to the bending vibration of heptazine rings. A series of peaks ranging between 1250 and 1630 $cm^{-1}$ were ascribed to the typical stretching modes of CN heterocycles. The broad absorption peak at around 3200 $cm^{-1}$ originated from the stretching vibration of the N-H bond [20]. For bare PANI, the peaks at 806 $cm^{-1}$ and 1132 $cm^{-1}$ were the out-of-plane and in-plane stretching vibration of the C-H bond. The peak at 1302 $cm^{-1}$ corresponded to the stretching vibration of the C-N bond. The peaks at 1485 $cm^{-1}$ and 1568 $cm^{-1}$ were ascribed to the vibration peaks of benzene ring and quinone ring, respectively [21–25]. For the CN/PANI heterojunction film, the characteristic peaks of CN and PANI were simultaneously observed.

The X-ray diffraction (XRD) patterns of pristine CN, PANI, and CN/PANI heterojunction film are shown in Figure 2b. For the pristine CN, two strong diffraction peaks at 13.1° and 27.4° were observed, corresponding to the (100) and (002) crystal plane of graphitic carbon nitride (JCPDS 87-1526) [26]. For bare PANI, the diffraction peaks at 14.94°, 20.22°, and 25.52° were assigned to the (011), (020), and (200) crystal planes of the emerald salt (ES) form of polyaniline [27]. The XRD pattern of the CN/PANI heterojunction film showed characteristic peaks of CN and PANI. The FTIR and XRD result indicated the simultaneous presence of CN and PANI within the film, implying the successful fabrication of the

CN/PANI heterojunction film. X-ray photoelectron spectroscopy (XPS) measurements were conducted to identify the elemental state of as-synthesized CN/PANI heterojunction film.

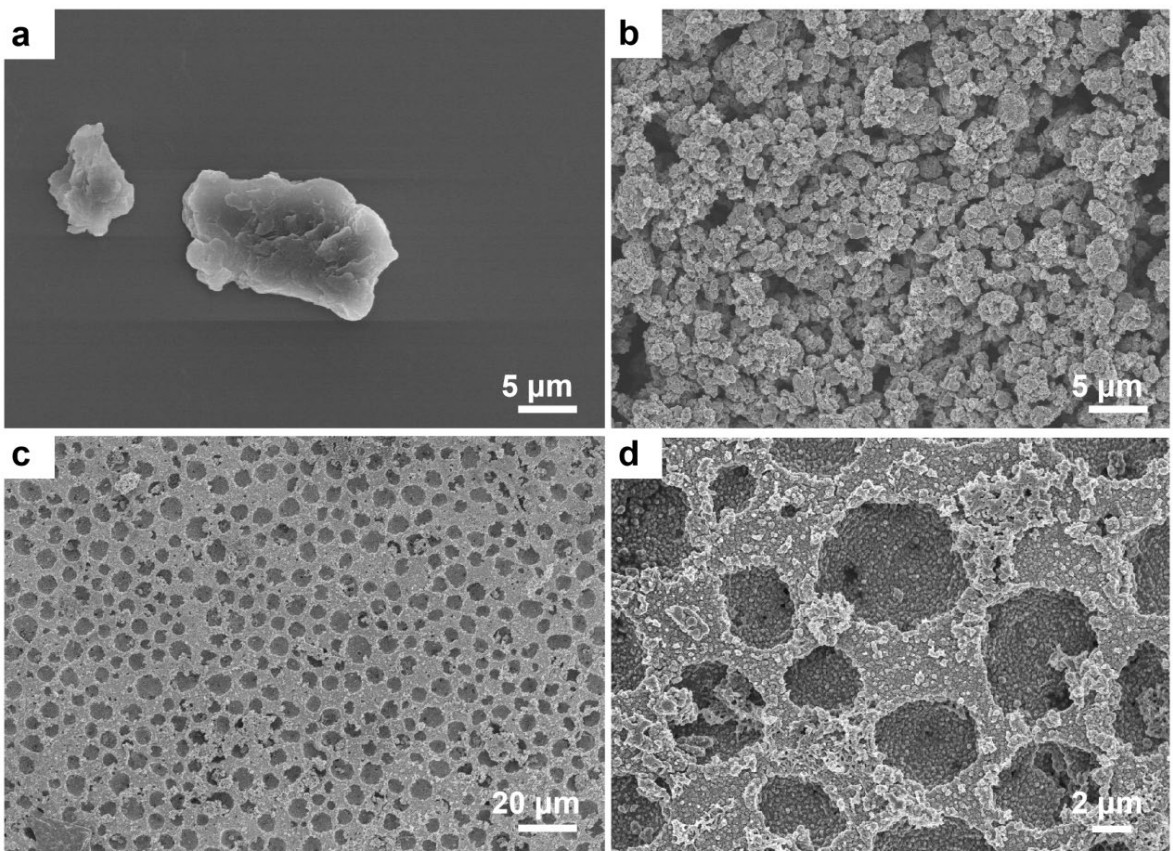

**Figure 1.** SEM images of (**a**) pristine CN, (**b**) pure PANI, and (**c**,**d**) CN/PANI heterojunction film.

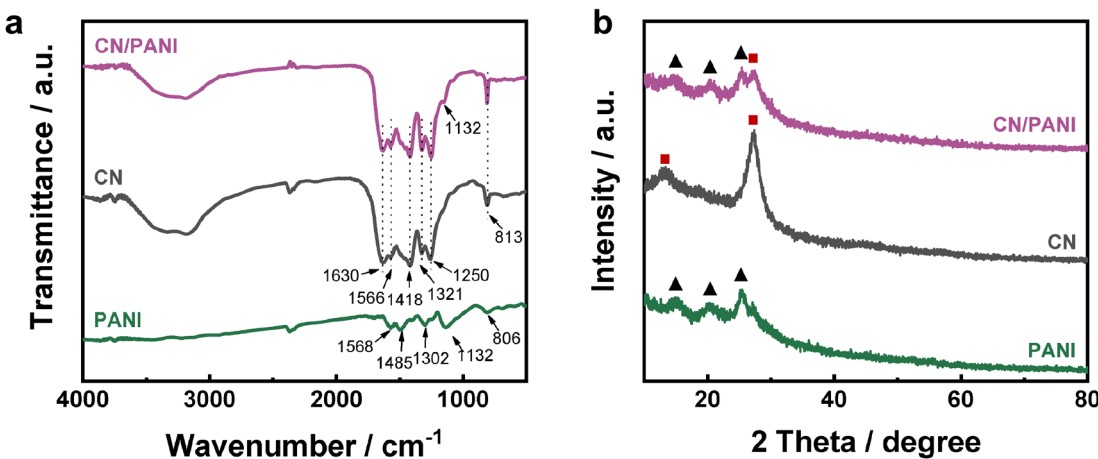

**Figure 2.** (**a**) FTIR spectra and (**b**) XRD patterns of pristine CN, PANI, and CN/PANI heterojunction film.

The survey XPS spectra of the CN/PANI heterojunction film in Figure 3a demonstrated the coexistence of C, N, and O elements. The high-resolution C 1s spectra deconvoluted into three peaks was exhibited in Figure 3b. The peaks centered at 288.4 and 284.8 eV were ascribed to N-C=N and C-C groups of CN [26]. The peak at binding energy of 286.4 eV was assigned to the C-N group of PANI [21]. The high-resolution N 1s spectra also confirmed the simultaneous presence of CN and PANI. The N 1s spectra showed in Figure 3c could be deconvoluted into four peaks centered at 398.6, 399.2, 400.1, and 401.1 eV, respectively.

The peaks at 398.6, 400.1, and 401.1 eV were associated with C-N=C, N-(C)$_3$, and -NHx groups of CN, while the peak at 399.2 eV was ascribed to C-N group of PANI [21,26]. The high-resolution O 1s spectra is given in Figure 3d. The peak at 532.2 eV was assigned to surface-adsorbed oxygen species of CN [20]. The XPS result further confirmed the successful construction of the CN/PANI heterojunction film.

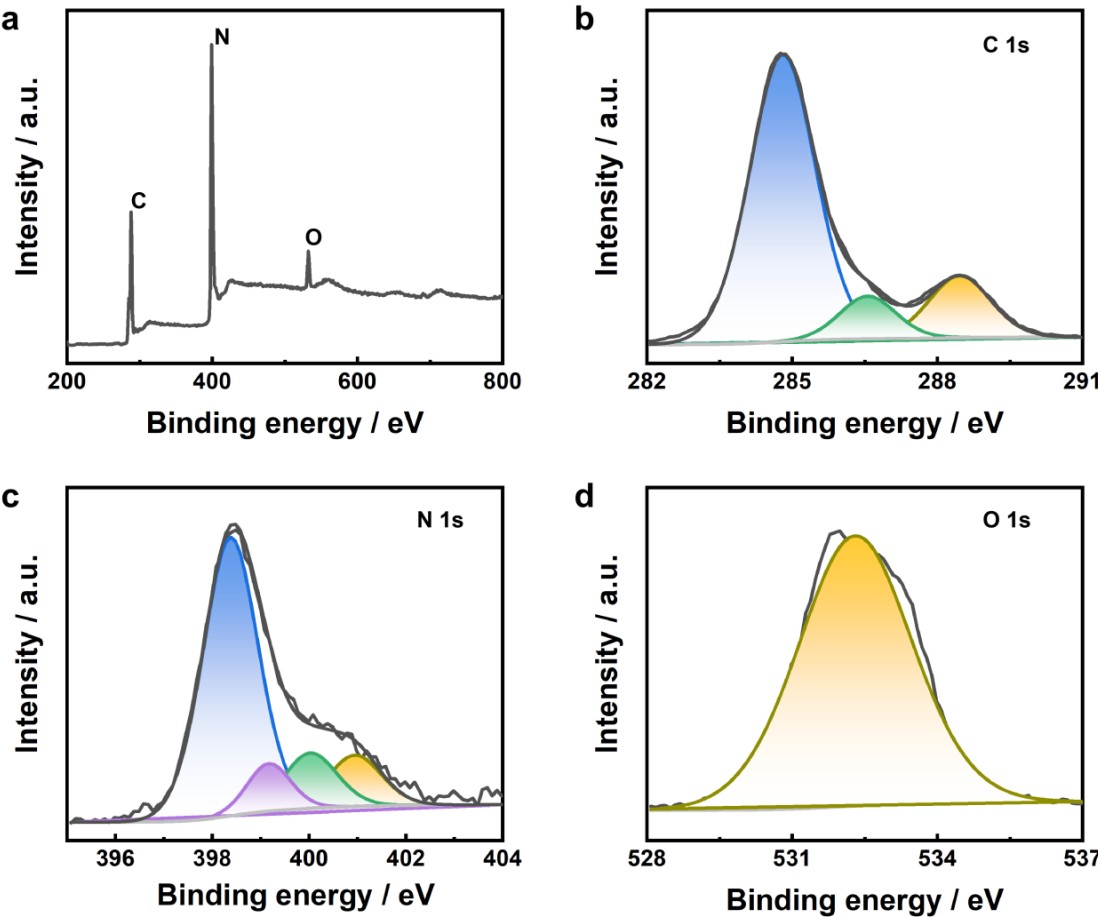

**Figure 3.** (**a**) Survey XPS spectra and (**b**–**d**) high-resolution XPS spectra of the CN/PANI heterojunction film: (**b**) C 1s, (**c**) N 1s, and (**d**) O 1s.

### 2.2. Photocatalytic Performance Evaluation

The photocatalytic H$_2$ evolution rate of the pristine CN, bare PANI, and CN/PANI heterojunction film were evaluated. As shown in Figure 4a, the bare PANI showed a minimum H$_2$ generation rate of 308.2 µmol·h$^{-1}$·g$^{-1}$, and the pristine CN exhibited an H$_2$ production rate of 569.1 µmol·h$^{-1}$·g$^{-1}$. The as-prepared CN/PANI heterojunction film showed a significantly enhanced H$_2$ evolution rate (3164.3 µmol·h$^{-1}$·g$^{-1}$) compared to the single component, indicating that the construction of the heterojunction film was favorable to boost the photocatalytic performance. The photocatalytic H$_2$ generraction activity of the CN/PANI heterojunction powder was measured for comparison (Figure S2). Its low H$_2$ evolution rate (1674.5 µmol·h$^{-1}$·g$^{-1}$) revealed that the the porous membrane fabrication was crucial to facilitate photocatalytic H$_2$ production. The photocatalytic H$_2$ evolution acticities of other similar CN-based and PANI-based heterojunctions were exhibited in Table S1 [28–33], suggesting the superior photocatalytic H$_2$ generation performance of the as-synthesized porous hydrophilic CN/PANI heterojunction film. The photocatalytic recycling tests of the CN/PANI heterojunction film was depicted in Figure 4b. Its photocatalytic H$_2$ generation activity was preserved after four cycles, confirming the excellent photostability of the as-prepared CN/PANI heterojunction film. To understand the enhanced photocatalytic performance of the CN/PANI heterojunction film, the surface

wettability, porosity characteristics, and optical and photoelectric conversion properties were characterized.

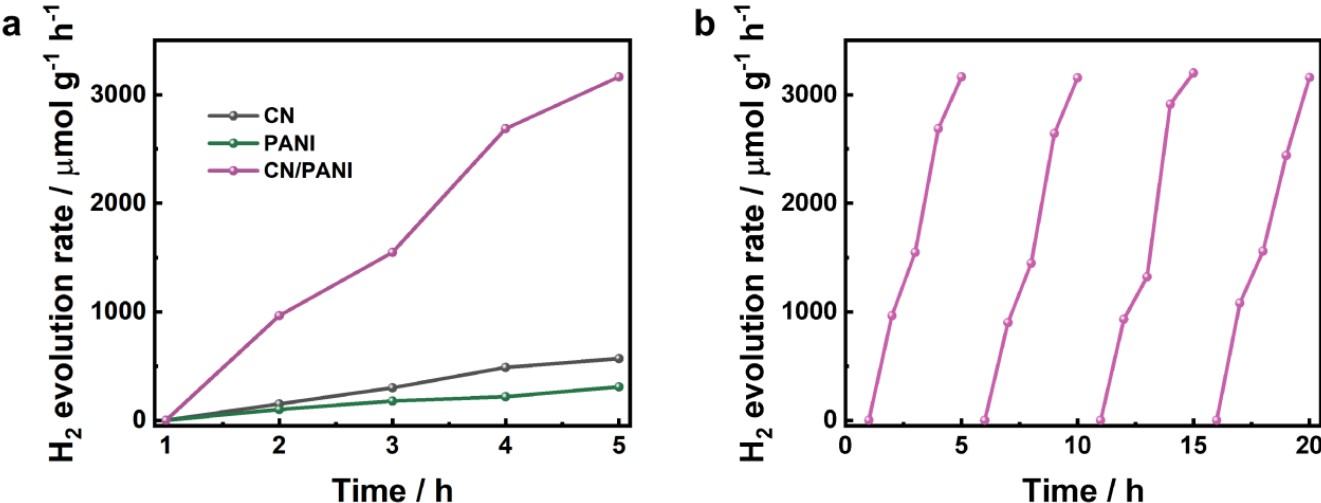

**Figure 4.** (**a**) Photocatalytic $H_2$ evolution rate of pristine CN, bare PANI, and the CN/PANI heterojunction film. (**b**) Cyclic $H_2$ production tests of the CN/PANI heterojunction film.

### 2.3. Mechanism Analysis

Figure 5 presents the $N_2$ adsorption/desorption isotherms of the pristine CN, PANI, and CN/PANI heterojunction film. The BET surface area, pore size, and pore volume of the as-prepared photocatalysts were displayed in Table 1. It could be seen that all of the photocatalysts show the typical type IV isotherms with a H3 hysteresis loop, proving their predominant mesoporous structures [34]. The measured BET surface areas of the pristine CN and PANI were only 9.47 and 5.75 $m^2 \cdot g^{-1}$, much lower than the fabricated CN/PANI heterojunction film (137.87 $m^2 \cdot g^{-1}$). The higher surface area of the CN/PANI film provided more catalytic active sites and a shorted diffusion distance for charge carrier, which was one reason for its improved photocatalytic activity.

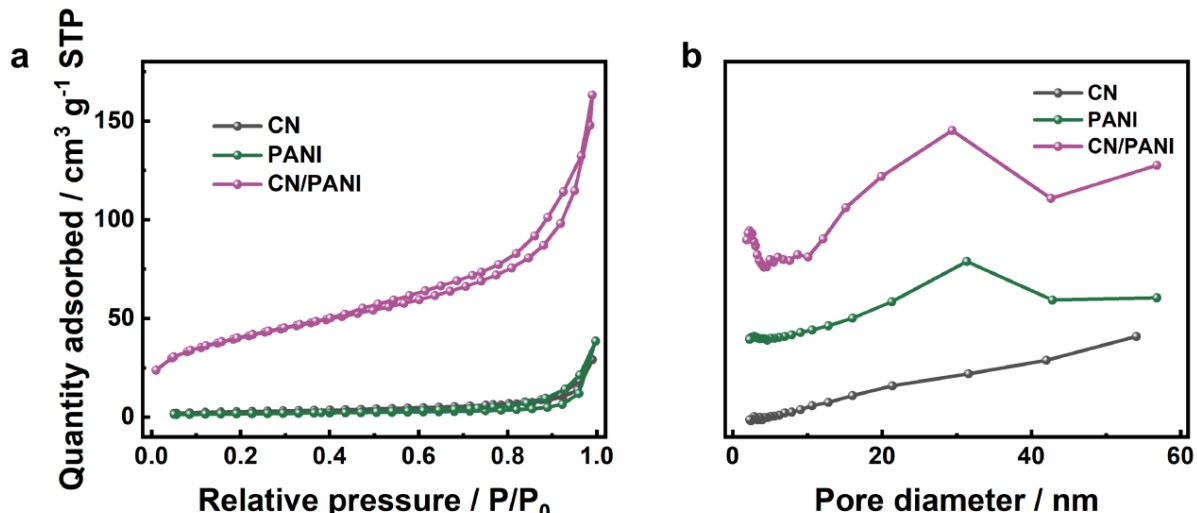

**Figure 5.** (**a**) $N_2$ adsorption–desorption isotherms of the pristine CN, PANI, and CN/PANI heterojunction film. (**b**) The corresponding pore size distributions calculated from BJH method.

The surface wettability of pristine CN, bare PANI, and CN/PANI heterojunction film were tested via the sessile drop method. As shown in Figure 6, the contact angle of the CN/PANI heterojunction film (35.0°) was evidently smaller than that of pristine CN photocatalyst (50.2°), indicating the improved hydrophilicity of the obtained heterojunction film. The enhanced surface wettability of the CN/PANI heterojunction film was beneficial

for the adsorption of water molecules and the escape promotion of gaseous product ($H_2$) [35], which was desirable for photocatalytic $H_2$ evolution enhancement.

**Table 1.** BET surface area, pore size, and pore volume of the pristine CN, PANI, and CN/PANI heterojunction film.

| Photocatalyst | BET Surface Area ($m^2 \cdot g^{-1}$) | Pore Size (nm) | Pore Volume ($cm^3 \cdot g^{-1}$) |
|---|---|---|---|
| CN | 9.47 | 9.6 | 0.057 |
| PANI | 5.75 | 15.9 | 0.045 |
| CN/PANI heterojunction film | 137.87 | 9.4 | 0.177 |

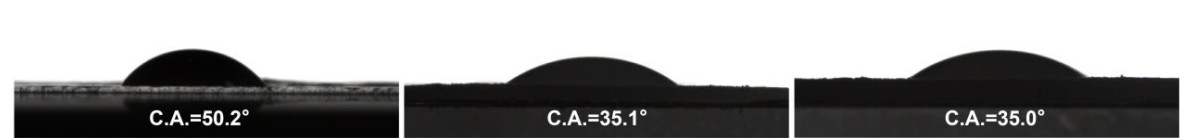

**Figure 6.** Water-droplet contact angles of (**a**) the pristine CN, (**b**) bare PANI, and (**c**) CN/PANI heterojunction film.

The visible light response capacity and photoelectric conversion efficiency of the as-prepared photocatalysts were characterized by the UV-vis diffuse reflectance spectra, PL spectra, and electrochemical measurements. The pristine CN showed an adsorption band edge of 460 nm due to the typical semiconductor band–band transition (Figure 7a) [28]. Bare PANI exhibited strong light adsorption in both the ultraviolet and visible spectrum regions attributed to the $\pi \rightarrow \pi^*$ transition [36]. The CN/PANI heterojunction film exhibited an overall spectrum response, which was conducive to utilize more photons to generate charge carriers for photocatalytic reaction.

The separation efficiency of photogenerated charge carriers of the as-synthesized photocatalysts was characterized by the intensity of PL emission spectra. As illustrated in Figure 7b, the pristine CN showed the highest PL emission intensity due to the high recombination rate of photoexcited charge carriers. Weak photoluminescence effect was detected for the bare PANI. In comparison with pristine CN, the PL emission intensity of the CN/PANI heterojunction film dramatically decreased, demonstrating the supriority of porous solid-state membrane fabrication on the separation promotion of photoexcited charge carriers. The migration resistance of photogenerated charge carriers for the as-prepared photocatalysts was measured by EIS spectra. The typical $\pi$-conjugated skeleton structure of conductive polymers (CPs) enabled good conductivity of the PANI molecular chain. Therefore, the CN/PANI heterojunction film showed significantly decreased slop of Nyquist plots in comparison with pristine CN (Figure 7c), implying the reduced transportation resistance at the interface [37]. The photocurrent density indicated the migration efficiency of photoinduced charge carriers of the as-prepared photocatalysts, which was displayed in Figure 7d. The pristine CN showed the lowest photocurrent density attributed to the high transportation resistance of charge carriers induced by its inherent $\Pi$-deficient conjugated system [26]. The CN/PANI film showed evidently increased photocurrent density, suggesting the heterojunction construction effectively promoted the migration of photogenerated charge carriers. The optical and electrochemical measurement revealed the extended visible light harvesting ability and optimized photoelectric conversion efficiency of the as-prepared CN/PANI heterojunction film, which was responsible for its improved photocatalytic $H_2$ generation performance.

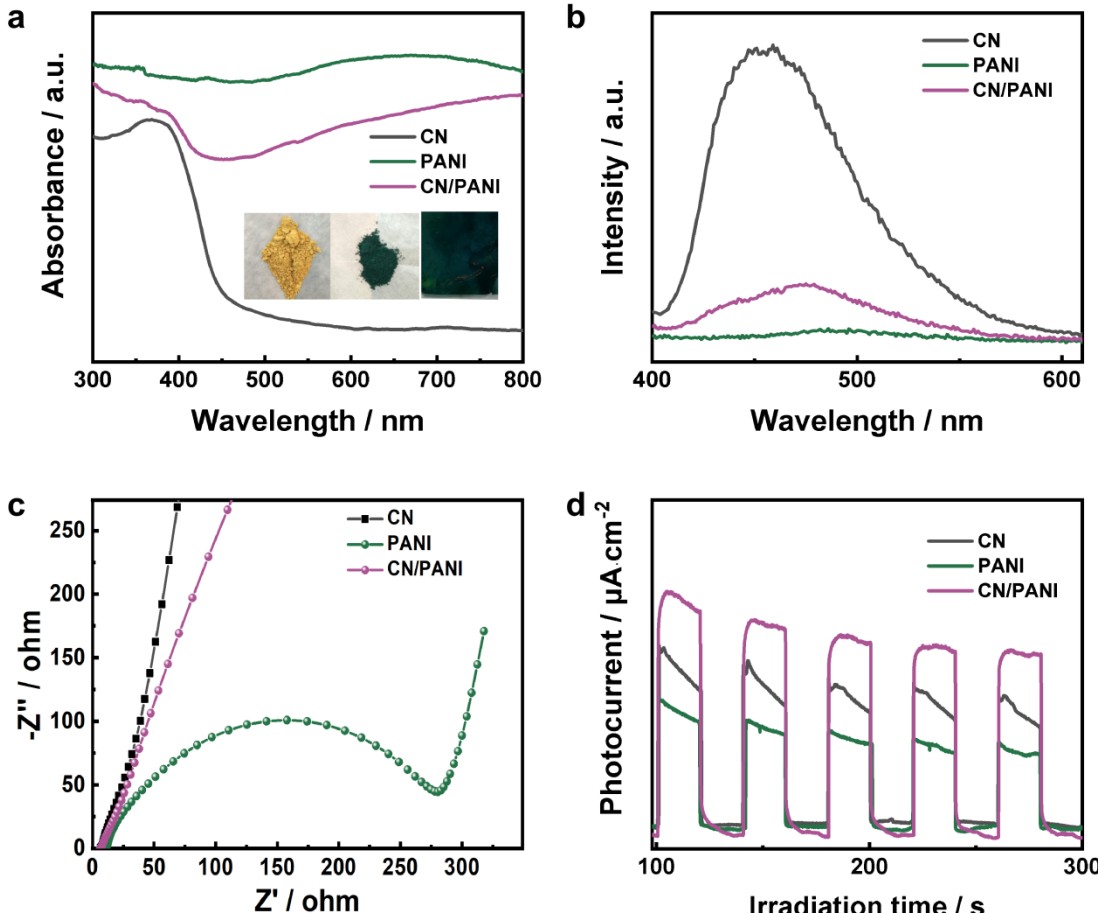

**Figure 7.** (**a**) UV-Vis diffuse reflectance spectra and optical pictures (inserted ones); (**b**) PL emission spectra; (**c**) EIS spectra; and (**d**) periodic on/off photocurrent response for the pristine CN, PANI, and CN/PANI heterojunction film.

The energy band edge potentials of CN and PANI were computed to elucidate the enhancement of the photocatalytic activity enhancement mechanism. The highest occupied molecular orbital (HOMO) and lowest unoccupied molecular orbital (LUMO) of PANI were composed of bonded π orbitals and anti-bonded π* orbitals. It was known that the HOMO and LUMO of PANI were 0.62 and −2.14 V, respectively. The band gap between orbitals was approximately 2.76 eV [14]. The band gap and conduction band minimum (CBM) potential of CN were determined to be 2.59 eV and −1.8 V according to its Tauc plot and Mott–Schottky plot (Figures S3 and S4). Then, the valence band minimum (VBM) potential was derived to be 0.79 V. The schematic illustration for the energy-band structure of the CN/PANI film was displayed in Figure 8.

The mechanism for improved photocatalytic H$_2$ production of the CN/PANI heterojunction film was proposed. After interfacial polymerication of aniline, a tight interface was formed between CN and PANI. Due to the relatively low CB and VB edge potential of CN, a type-II heterostructure was constructed and a charge carriers channel was generated at the interface of two phases. Under visible light irradiation, PANI absorbed photons to induce a π→π* transition, and the excited electrons were transferred to the π* orbital. At the same time, CN could be excited and thus create photogenerated electronhole pairs. Following the standard type II heterojunction transfer path of photogenerated carriers, the electrons on the LUMO of PANI were able to inject into the CB of CN and the holes on the VB of CN could transfer to the HOMO of PANI, which effectively promoted the separation and directional transportation of photogenerated charge carriers. The electrons accumulated on the CB of CN participate in the water-reduction reaction, and the holes accumulated on the

HOMO of PANI take part in the TEOA oxidation reaction. Consequently, the CN/PANI heterojunction film showed enhanced photocatalytic $H_2$ generation performance attributed to increased surface area, improved surface wettability, an extended visible light response, and the separation and migration promotion of photoexcited charge carriers.

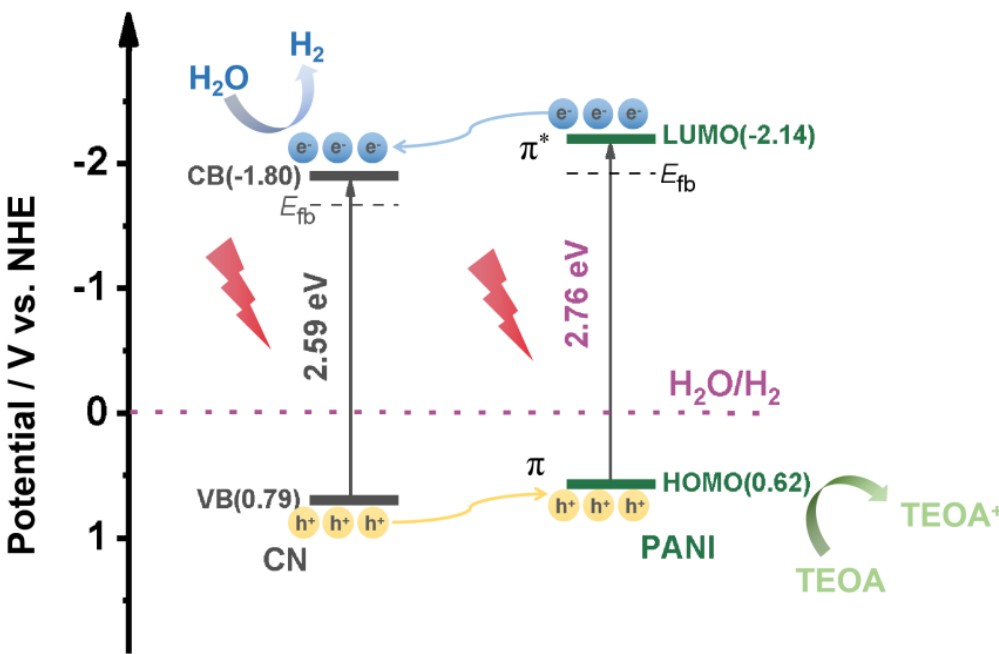

**Figure 8.** The schematic illustration for electronic energy-band structure and photocatalytic $H_2$ evolution improvement mechanism of CN/PANI heterojunction film under visible light irradiation.

## 3. Materials and Methods

### 3.1. Chemicals

Urea ($CH_4N_2O$), sodium sulfate ($Na_2SO_4$), and Nafion were purchased from Beijing Chemical Works, Beijing, China. Aniline monomer (An), hydrochloric acid (HCl), ammonium persulfate (($NH_4)_2S_2O_8$, APS), and ethanol were supplied by Yongda Chemical Reagent, Tianjin, China. All of the chemicals were analytic-grade and used without purification. Fluoride tin oxide (FTO) glass was purchased from Xiangcheng Technology, Hunan, China. Deionized water (18.2 $M\Omega \cdot cm$) was used throughout all of the experiments.

### 3.2. Photocatalysts Preparation Procedure

#### 3.2.1. Synthesis of CN

The pristine graphitic carbon nitride (CN) was prepared by direct thermal polymerization using urea as starting material. Typically, 10 g of urea was put out in a 50 mL ceramic crucible with a cover and calcined at 520 °C for 2 h in a murfle furnace with a heating rate of 5 °C/min. The obtained yellow product was washed several times by water and ethanol and dried in an oven for further use.

#### 3.2.2. Synthesis of CN/PANI Composites and CN/PANI@PCL Film

The ordered porous polycaprolactam lactone (PCL) film substrate was prepared by the solvent evaporation self-organization method. The detailed process had been described in our previous study [38]. The CN/PANI@PCL composite films were prepared by impregnation adsorption of aniline (An) monomer and CN on PCL substrate, followed by in situ polymerization. Typically, 40 mL of the HCl solution containing 0.01 mol of An, the PCL film substrate (2.5 cm × 4 cm), and 0.5 g of CN was added to a reactor. The mixture was stirred in an ice-water bath for 1 h to obtain a homogeneous solution. A cooled HCl solution of APS was added dropwise into the homogeneous solution to initiate the in situ polymerization of aniline. After the complete polymerization, the film was washed

three times by water and ethanol and dried at 80 °C in vacuum oven for 12 h to obtain the CN/PANI composite loaded on the PCL substrate. For comparison, the CN/PANI composite was synthesized by the same procedure without adding a PCL film substrate.

### 3.3. Characterizations

The morphology of the as-prepared samples was observed on a Hitachi S4800 scanning electron microscopy (SEM). Fourier transformed infrared (FTIR) spectra were recorded using a Thermo Nicolet iS50 FTIR spectrometer. Powder X-ray diffraction (XRD) patterns were collected with Bruker D8 Advance diffractometer with Cu K$\alpha$ radiation at a scanning rate of 5°/min. Surface areas and pore size distributions of the photocatalysts were acquired by nitrogen physisorption at 77 K on a Nova 1200e Surface Area and Porosity Analyzer. X-ray photoelectron spectroscopy (XPS) measurements were performed on PHI Quantera II spectrometer using nonmonochromatized Al-K$\alpha$ X-ray as an excitation source. The water contact angle was measured with a JY-82B Kruss DSA video-based contact angle goniometer. The average CA value was obtained by measuring the sample at four different positions. UV-vis diffuse reflection spectra was recorded in the spectral region of 200–800 nm on a Shimadzu UV-2550 spectrophotometer with $BaSO_4$ as the reference substance. Room temperature photoluminescence (PL) spectra were monitored at a PerkinElmer LS-55 Luminesence Spectrometer under an excitation wavelength of 365 nm. Conductivity measurements were conducted on a Chenjing ET3000 Hall system.

### 3.4. Photoelectrochemical Tests

Photoelectrochemical tests were performed on a Chenhua CHI760E electrochemical workstation with a standard three-electrode photoelectrochemical cell. A gauze platinum, Ag/AgCl (saturated KCl), and FTO glass coated with the CN/PANI membrane were used as the counter electrode, reference electrode, and working electrode, respectively. The three electrodes were immersed in a sodium sulfate ($Na_2SO_4$) electrolyte solution (0.5 M, pH 6.8). A 300 W Xe lamp (Perfect Light PLS-SXE300, Beijing, China) equipped with a 420 nm UV-cutoff filter was used as irradiation light resource. The solution was continuously in an $N_2$-purged flow to remove $O_2$ before photoelectrochemical measurements. The working electrode was illuminated to record the electrochemical impedance spectra (EIS) under the perturbation signal of 5 mV and the frequency ranged from 0.1 Hz to 100 kHz. Periodic on/off light was applied to monitor the transient photocurrent response in 400 s.

### 3.5. Evaluation of Photocatalytic Performance

A 300 W Xe lamp (Perfect Light PLS-SXE300, Beijing, China) equipped with a 420 nm UV-cutoff filter was used as irradiation resource for photocatalytic $H_2$ evolution measurements. For the photocatalytic film system, four pieces of as-prepared film (2.5 cm × 4 cm) were immersed into 300 mL aqueous solution containing 10% TEOA and 3 wt.% Pt. Before light irradiation, the reactor was bubbled with $N_2$ to remove dissolved $O_2$. Then, the Xe lamp was turned on to initiate the photocatalytic $H_2$ generation test and the gaseous products were analyzed on a gas chromatography (Fuli GC9790II(PLF-01), Taizhou, China). The cyclic stability measurements of the CN/PANI film were evaluated under the same procedure.

## 4. Conclusions

In summary, we successfully synthesized the porous and hydrophilic CN/PANI solid-state film loaded on the porous PCL substrate. The formed type II CN/PANI heterojunction achieved an overall spectrum response and high-efficient separation and migration of photogenerated charge carriers. The porous morphology and hydrophilic surface contributed to an increased surface area and improved surface wettability, enhancing the adsorption and activation of reactants ($H_2O$). Meanwhile, the design of solid film was conducive to maintain the long-term stability of the photocatalyst. Thereby, the obtained porous hydrophilic CN/PANI heterojunction membrane showed the significantly enhanced

photocatalytic $H_2$ evolution performance. Such work will provide new ideas for further improving the feasibility of CN-based photocatalysts in the practical application of hydrogen energy production.

**Supplementary Materials:** The following supporting information can be downloaded at https://www.mdpi.com/article/10.3390/catal13010139/s1, Figure S1: The SEM image of pristine CN/PANI heterojunction film; Figure S2: Photocatalytic $H_2$ evolution rate of the CN/PANI heterojunction powder; Figure S3: Tauc plot of pristine CN. Figure S4: Mott–Schottky plot of pristine CN; Table S1: Comparison in photocatalytic H2 evolution activities of similar CN based heterojunctions.

**Author Contributions:** Conceptualization, X.Y. and Y.Z.; methodology, J.D. and R.J.; formal analysis, X.H.; investigation, X.Y.; data curation, X.Y. and Y.Z.; writing—original draft preparation, X.Y. and Y.Z.; writing—review and editing, Y.W. and R.J.; visualization, X.H.; supervision, Y.Z.; project administration, X.H.; and funding acquisition, X.Y. and R.J. All authors have read and agreed to the published version of the manuscript.

**Funding:** This research was funded by the Jilin Province Science and Technology Development Project (20220101248JC), the Jilin Provincial Education Department Project (JJKH20230538KJ), and the Doctoral Research Start-up Fund of Jilin Medical University (JYBS2021032LK).

**Data Availability Statement:** The data presented in this study are available on request from the corresponding author.

**Acknowledgments:** The authors would like to thank Shiyanjia Lab (www.shiyanjia.com, accessed on 1 January 2021) for their help on material characterizations.

**Conflicts of Interest:** The authors declare no conflict of interest.

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
