# Peer review of "Fabrication of Porous Hydrophilic CN/PANI Heterojunction Film for High-Efficiency Photocatalytic H2 Evolution"

_catalysts, doi:10.3390/catal13010139_

Round 1

Reviewer 1 Report

The manuscript "Fabrication of Porous hydrophilic CN/PANI Heterojunction 2 Film for High-efficiency Photocatalytic H2 Evolution" is systematically designed and presented. Different experimental parameters such as SEM, FTIR, XPS, BET, etc. are studied. and mechanism of photocatalytic H2 evolution has also been discussed with CN/PANI heterojunction and separation of charge carriers. However, these nanostructures are not new or novel. In general, this paper (catalysts-2106607) has provided some new insights into old synthesized CN/PANI composites. Therefore, I accept this manuscript as it is.

Author Response

We thank Referee 1 very much for the encouragement. No action is needed here.

Reviewer 2 Report

1.In the field of photocatalysis, the orbital energy of the semiconductor is very important, but in this paper, the author not only calculate the Eg. “UV-Vis diffuse reflectance spectra and optical pictures (inserted ones)” I cannot found the Eg value and related figure.

2. How about its recycle ability for the title samples?

3. For bare PANI, the peaks at 806 cm-1 and 1132 cm-1 were the out-of-plane and in-plane stretching vibration of the C-H bond. The peak at 1302 cm-1 corresponded to the stretching vibration of C-N bond. The peaks at 1485 cm-1 and 1568 cm-1 were ascribed to the vibration peaks of benzene ring and quinone ring, respectively. Some refs could be updated, such as ACS Appl. Mater. Interfaces., 2021, 13, 12463−12471; Micropor. Mesopor. Mat, 341(2022) 112098.Inorganics, 10(2022) 202 and CrystEngComm, 2017,19:4368-4377.

4. I want to ask the authors to check the morphology of pristine CN/PANI heterojunction film.

5. The authors should compare the similar materials on this function of photocatalytic H2 Evolution

Round 2

Reviewer 2 Report

accepted.